# BZR1 Physically Interacts with SPL9 to Regulate the Vegetative Phase Change and Cell Elongation in *Arabidopsis*

**DOI:** 10.3390/ijms221910415

**Published:** 2021-09-27

**Authors:** Lingyan Wang, Ping Yu, Jinyang Lyu, Yanfei Hu, Chao Han, Ming-Yi Bai, Min Fan

**Affiliations:** The Key Laboratory of Plant Development and Environmental Adaptation Biology, Ministry of Education, School of Life Science, Shandong University, Qingdao 266237, China; wly605240639@163.com (L.W.); 18463758489@163.com (P.Y.); lvjy890727@sdu.edu.cn (J.L.); huyf199905@163.com (Y.H.); hanchao@sdu.edu.cn (C.H.); baimingyi@sdu.edu.cn (M.-Y.B.)

**Keywords:** vegetative phase change, cell elongation, *SPL9*, BZR1

## Abstract

As sessile organisms, the precise development phase transitions are very important for the success of plant adaptability, survival and reproduction. The transition from juvenile to the adult phase—referred to as the vegetative phase change—is significantly influenced by numbers of endogenous and environmental signals. Here, we showed that brassinosteroid (BR), a major growth-promoting steroid hormone, positively regulates the vegetative phase change in *Arabidopsis thaliana*. The BR-deficient mutant *det2-1* and BR-insensitive mutant *bri1-301* displayed the increased ratio of leaf width to length and reduced blade base angle. The plant specific transcription factors SQUAMOSA PROMOTER BINDING PROTEIN-LIKE (SPL) are key masters for the vegetative phase transition in plants. The expression levels of *SPL9*, *SPL10* and *SPL15* were significantly induced by BR treatment, but reduced in *bri1-116* mutant compared to wild-type plants. The gain-of-function *pSPL9:rSPL9* transgenic plants displayed the BR hypersensitivity on hypocotyl elongation and partially suppressed the delayed vegetative phase change of *det2-1* and *bri1-301*. Furthermore, we showed that BRASSINAZOLE-RESISTANT 1 (BZR1), the master transcription factor of BR signaling pathway, interacted with *SPL9* to cooperatively regulate the expression of downstream genes. Our findings reveal an important role for BRs in promoting vegetative phase transition through regulating the activity of *SPL9* at transcriptional and post-transcriptional levels.

## 1. Introduction

The transition from the juvenile to adult stage is referred to as the vegetative phase change and is crucial for the reproductive success and survival of higher plants [1]. In Arabidopsis, the change of vegetative stage can be characterized by the appearance of abaxial trichomes, increasing the complexity of leaf shape and the ratio of leaf length to width [2,3]. The microRNA 156 (miR156) and its target transcription factors SQUAMOSA PROMOTER BINDING PROTEIN-LIKE (SPL) are the main regulatory factors of the vegetative phase transition in plants [4,5]. A wide range of signals, including sugar, gibberellin acid (GA), abscisic acid (ABA), auxin and endogenous epigenetic factors, regulate the activity of the miR156-SPL module to modulate the vegetative phase change [3,6,7,8]. These interconnected signal transduction networks integrate multiple developmental, environmental and hormonal signals to precisely optimize the vegetative developmental phase transition in the changing environmental conditions.

MicroRNA156 (miR156) is an evolutionarily conserved miRNA, which determines the juvenile-to-adult transition and regulates diverse aspects of plant growth and development [9,10,11,12]. MiR156 is enriched in the juvenile-stage leaves and gradually declines as the shoot develops, while the expression levels of *SPLs* are reversed, which are abundant at adult stages [9,10,13,14]. MiR156 is encoded by eight homologous genes, *miR156A-miR156H* in the Arabidopsis genome [5]. The single *mir156a* and *mir156c* mutant displayed the slightly early vegetative phase change, but the *mir156a/c* double mutant and *mir156a/b/c* triple mutant exhibited much early juvenile-to-adult transition [15,16]. Constitutive overexpression of *miR156* in Arabidopsis, as well as in maize, rice, tomato, and tobacco, all led to a delay in the phase transition from juvenile to adult [2,10,14,16,17,18]. However, overexpression of *miR156* target mimic, *MIM156*, which produces an mRNA specifically binding to miR156, inhibits the activity of miR156 and increases the transcript levels of *SPLs* (such as *SPL3*, *SPL4*, *SPL5*, *SPL9*, and *SPL15*), thus shortening the juvenile phase [9,10,19,20]. *SPLs* encode a family of plant specific transcription factors, which contain a highly conserved DNA-binding domain, the SBP domain [21,22,23]. In Arabidopsis, 10 of the 16 *SPL* genes have a sequence complementary to miR156, and they differently respond to the changes of miR156 levels [5,22,23,24,25,26]. MiR156 blocks the expression of all 10 targeted *SPLs* in the first two rosette leaves, which is due to the high levels of *miR156* in these leaves. Because the transcript levels of *miR156* decrease in the subsequent leaves, the transcripts of *SPL3*/*SPL9*/*SPL15* quickly increase [5,23,26]. SPL9 and SPL15 are the most well-understood members of SPL family in *Arabidopsis thaliana.* The loss-of-function mutants *spl9* and *spl15* results in the increase of shoot branching, shortening plastochron during vegetative growth and delay of the phase transition from juvenile to adult. Whereas the transgenic plants *proSPL9:rSPL9*, which expressed a miR156-resistant form of *SPL9* driven by its native promoter, displayed the reduced shoot branch and much early vegetative phase change [10,18,27,28]. SPL9 directly binds to the promoter of *LAS* to repress axillary bud formation [29]. Our recent study further showed that gibberellin inhibited the formation of axillary buds in Arabidopsis by promoting the degradation of DELLA proteins to release their inhibiting effects on the transcriptional activity of SPL9 [30].

Brassinosteroids (BRs), as a classic of plant steroid hormone, play important roles in plant growth and development [31,32]. BRs directly bind to the extracellular domain of the receptor kinase BRASSINOSTEROID INSENSITIVE1 (BRI1) which interacts with coreceptor BRI1-ASSOCIATED RECEPTOR KINASE1 (BAK1), and as a result rapidly activates the receptor’s intracellular kinase domain through homology dimerization and trans-phosphorylation [33,34,35,36]. Activated BRI1 phosphorylates activate a series of kinases, such as BRASSINOSTEROID-SIGNALING KINASE1 (BSK1) and CONSTITUTIVE DIFFERENTIAL GROWTH1 (CDG1), as well as the Ser/Thr phosphatase BRI1-SUPPRESSOR1 (BSU1) which inactivates the kinase BRASSINOSTEROID INSENSITIVE2 (BIN2) by dephosphorylation, then BIN2 was degraded by the F-box protein KINKSUPPRESSED (KIB1) [37,38,39,40]. Inactivation of BIN2 makes the unphosphorylated transcription factors BRASSINAZOLE RESISTANT 1 (BZR1) and BRI1-EMS-SUPPRESSOR 1 (BES1) by PROTEIN PHOSPHATASE 2A (PP2A) enter and accumulate in the nucleus, which directly regulates the expression of BR responsive genes [41,42,43,44,45,46]. BZR1 and BES1 interact with other growth promoting transcription factors, such as PIF4 and ARF6, to regulate cell elongation by cooperatively regulating their downstream gene expression [47,48]. Hydrogen peroxide (H_2_O_2_) induces the oxidation of BZR1 to increase the transcription activity of BZR1 by enhancing the interaction between BZR1 with PIF4 and ARF6 [49]. BZR1 and BES1 also have been reported to interact with diverse proteins to integrate with BR and numbers of hormonal and environmental signals to regulate plant growth and development.

Genetic analysis showed that the BR-deficient and BR-insensitive mutants displayed the round leaves and the significant decreased ratio of leaf length to width compared to wild-type plants, suggesting BR maybe involved in the vegetative phase change. Here, in this study, we showed that BR treatment promoted the expression of *SPL9*, *SPL10*, and *SPL15*, while mutation of *BRI1* led to the decreased expression levels of these genes. The *pSPL9:rSPL9* displayed the BR hypersensitivity and suppressed the delayed vegetative phase change of *det2-1* and *bri1-301*. BZR1 interacts with SPL9 to coordinately regulate the downstream gene expression. These results demonstrate that BR promotes the vegetative growth of plants by increasing the activity of SPL9 at both transcriptional and post-transcription levels.

## 2. Results

### 2.1. BZR1 Interacts with SPL9 In Vitro and In Vivo

To further investigate the function of BZR1 in plant growth and development, we screened the proteins interacting with BZR1 by Yeast two-hybrid (Y2H) system. SPL9 was identified from the putative BZR1-interacting candidates. Additional Y2H assays showed that SPL9 interacted with BZR1 and BES1 in yeast (Figure 1A). BZR1 contains the amino-terminal DNA binding domain, PEST domain, which is responsible for the interaction with PP2A, and the carboxy-terminal domain, which itself is critical for the association with BIN2. The Y2H assays showed that SPL9 had the high binding ability with BZR1-C, weak binding ability with PEST domain, and did not interact with BZR1-N (Figure 1B,C). To confirm the interaction between BZR1 and SPL9, protein-protein pull down assay was performed using purified proteins fused glutathione S transferase (GST) or maltose binding protein (MBP) expressed from *Escherichia coli*. The results showed that GST-SPL9 protein pulled down MBP-fused BZR1, but not MBP alone (Figure 1D). To determine whether BZR1 interacts with SPL9 in plants, transient bimolecular fluorescence complementation (BiFC) assays were carried out in the tobacco leaves. The results showed that the strong fluorescent signals were detected in the nucleus of the epidermal cells of tobacco leaves when BZR1-cYFP was co-transformed with SPL9-nYFP, but not with control x-nYFP (Figure 1E). Furthermore, we performed Co-IP assay using YFP-trap beads and Arabidopsis protoplast, in which co-expressing 35S:GFP and 35S:*bzr1-1D*-MYC or co-expressing 35S:SPL9-GFP and 35S:*bzr1-1D*-MYC. The results showed that BZR1 interacted with SPL9 in plants (Figure 1F). All these results proved that BZR1 interacts with SPL9 in vitro and in vivo.

### 2.2. Enhancing the Activity of SPL9 Led to the BR Hypersensitivity

In Arabidopsis, several members of the SPL family are regulated post-transcriptionally by miR156. To determine whether the miR156 and SPL9 are involved in the BR signaling pathway, we studied the BR response of wild type, *miR156B-Ox*, in which *miR156B* was expressly driven by *35S* promoter, the miR156 target mimics line (*MIM156*) that reduces miR156 activity, and *pSPL9:rSPL9* plants where the resistant SPL9 (rSPL9) nontargeted by miR156 was expressed under its native promoter. The results showed that BR promoted the hypocotyl elongation of wild-type plants in a dose-dependent manner under the consistent light condition. The *miR156B-Ox* and *spl9*/*spl15* displayed the BR response similar to that of wild-type plants, but the *MIM156* and *pSPL9:rSPL9* displayed hypersensitivity to BR compared to wild-type plants (Figure 2A–C and Appendix A). To determine the BR response of *miR156/SPL9* module, we grew wild type, *miR156B-Ox* and *pSPL9:rSPL9* on the ½ MS medium containing different concentrations of BR biosynthesis inhibitor propiconazole (PPZ). We found *miR156B-Ox* displayed much smaller rosette leaves than wild-type plants, and *pSPL9:rSPL9* showed significantly increased rosette leaf sizes (Appendix A). Furthermore, the *bzr1-1D* gain-of-function mutant displayed the PPZ-resistant phenotypes, but such PPZ resistance of *bzr1-1D* were significantly reduced by *miR156* overexpression (Appendix A). These results indicated that the plants with high activity of SPL9 are hypersensitive to BR.

### 2.3. rSPL9 Partially Suppressed the Dwarf Phenotypes of det2-1 and bri1-301

To further investigate the functions of SPL9 in BR signaling pathway, we crossed *pSPL9:rSPL9* with BR-deficient mutant *det2-1* and BR-insensitive mutant *bri1-301* to generate *pSPL9:rSPL9/det2-1* and *pSPL9:rSPL9/bri1-301*. When growing in light, *det2-1*, and *bri1-301* seedlings had shorter hypocotyl length and the hypocotyl length of *pSPL9:rSPL9* is similar to that of Col-0, but *pSPL9:rSPL9* can partially suppress the short hypocotyl phenotypes of *det2-1* and *bri1-301* (Figure 3A,B). When growing in the dark, *pSPL9:rSPL9* plants still can partially suppress the short hypocotyl phenotype of *det2-1* (Figure 3C,D). The results indicated that SPL9 is a positive regulator of BR signaling pathway.

### 2.4. BR Ppromotes the Change of Vegetative Phase in Arabidopsis

The miR156-SPL9 module has been reported to play a critical role in the progression from the juvenile phase to the adult phase in plants. Considering that BZR1 interacts with SPL9 to regulate cell elongation, we want to determine whether BR and BZR1 participate in the vegetative phase change in plants. The BR-deficient mutants *det2-1*, and BR-insensitive mutants *bri1-301*, and the constitutively active forms *bin2-1* all displayed the increased ratio of leaf width to length, and the decreased blade base angle (Figure 4A–C). Meanwhile the BR signal enhanced materials such as *BRI1* overexpression (*BRI1-Ox*), *DWF4* overexpression (*DWF4-Ox*), and *bzr1-1D* overexpression (*bzr1-1D-Ox*), showed the decreased ratio of leaf width to length and the increased blade base angle (Figure 4A–C). These results indicated that BR promotes the vegetative phase change.

To determine the role of SPL9 in BR-mediated vegetative phase change, we described the growth phenotypes of wild type, *det2-1*, *bri1-301*, *pSPL9:rSPL9*, *pSPL9:rSPL9*/*det2-1* and *pSPL9:rSPL9*/*bri1-301*. The results showed of *det2-1* and *bri1-301* have a bigger ratio of leaf width to length and smaller blade base angle, while *pSPL9:rSPL9* have a smaller ratio of leaf width to length and bigger blade base angle (Figure 5A–F, Appendix A). What’s more, the rounder leaves of *det2-1* and *bri1-301* were suppressed by *pSPL9:rSPL9* (Figure 5C, Appendix A). In addition, the *pSPL9:rSPL9* can partially affect the rosette leaf numbers and leaf growth rate of *det2-1* and *bri1-301* (Figure 5B,D, Appendix A). Furthermore, we found that Col-0 appeared the abaxial trichome in the 7th rosette leaf, while in *bri1-301* there appeared the abaxial trichome in the 9th leaf, significantly delaying the appearance of abaxial trichomes in the *pSPL9:rSPL9* plant (Appendix A). These results indicated that SPL9 promotes vegetative phase change downstream of BR.

### 2.5. BR Induces the Expression of Several SPL Genes

Given the important roles of BR on the progression from the juvenile phase to the adult phase, we speculated that BR regulate the expression of *miR156* and *SPL* genes. To test this hypothesis, we analyzed the expression levels of *SPL* genes in wild type and BR-insensitive mutant *bri1-116*. Quantitative RT-PCR analysis showed that the expression levels of *SPL9*, *SPL10* and *SPL15* were significantly reduced in *bri1-116* mutant (Figure 6A). To further analyze the effects of BR on the expression of *SPL* genes, we analyzed the transcript levels of *SPL* genes in wild type with or without BR treatment. The results showed that BR treatment induced the expression of *SPL9*, but had no significant effects on the expression of *SPL10* and *SPL15* in wild type plants (Figure 6B). In order to eliminate the effects of miR156 on the expression of *SPL* genes, we analyzed the regulation of BR on the expression of *SPLs* in *MIM156* plants. The expression levels of *SPL9*, *SPL10* and *SPL15* were significantly increased in *MIM156* compared to that in wild-type plants (Figure 6B). These results revealed that BR induces the expression of *SPL9*, *SPL10* and *SPL15*.

### 2.6. BZR1 and SPL9 Coordinately Regulate the Expression of Downstream Genes

The HLH transcription factors PRE promote cell elongation participating in various hormones and environmental signals. Our previous study showed that BZR1 directly binds to the promoters of *PREs* to induce their expression. To examine whether SPL9 promotes cell elongation by regulating the expression of *PREs*, we performed the quantitative RT-PCR analysis in wild type, *det2-1*, *spl9/spl15*, *pSPL9:rSPL9* and *pSPL9:rSPL9/det2-1.* The results showed the expression levels of *PRE1*, *PRE5* and *PRE6* were similar in wild type and *spl9*/*spl15*, but much higher in *pSPL9:rSPL9* lines than that in wild-type plants, suggesting SPL9 induces the expression of *PRE1*, *PRE5* and *PRE6* (Figure 7A and Appendix A). Consistent with previous results, the transcript levels of *PRE1*, *PRE5* and *PRE6* were decreased in *det2-1* mutants, but these decreased expression levels of these genes were partially suppressed by *pSPL9:rSPL9* (Figure 7A). To test whether BZR1 and SPL9 cooperatively regulate the expression of *PREs*, we performed transient gene expression analysis by generating promoter-luciferase (LUC) reporter construct with *PRE5* promoter in mesophyll protoplasts of Arabidopsis leaves. We observed that the luciferase activity derived from the *pPRE5:LUC* increased when BZR1 and SPL9 were transfected alone, and significantly induced when BZR1 and SPL9 were co-expressed (Figure 7B). These results indicated that BZR1 and SPL9 cooperatively induce the expression of *PRE5*.

## 3. Discussion

Plant vegetative phase change is regulated by various of environmental and endogenous signals largely through influencing the activity of miR156-SPL module. Here, we showed that BR play critical roles for the transition from the juvenile to adult stage. BR-deficient mutant *det2-1* and BR-insensitive mutant *bri1-301* both displayed the round leaves and the increased ratio of leaf width to length, while overexpression of *BRI1* or *DWF4* exhibited the decreased ratio of leaf width to length. BR treatment significantly increased the expression of *SPL9*. Activated *SPL9* by overexpression of *MIM156* or miR156-resistant form *rSPL9* resulted in the hypersensitivity to BR and partially suppressed the dwarf phenotypes of *det2-1* and *bri1-301*. Furthermore, we showed that BZR1 interacts with SPL9 to coordinately regulate the expression of downstream genes. Our work reveals that age and BR pathways are integrated to regulate plant growth and development through the direct physical interaction between SPL9 and BZR1.

BRs function as one type of growth-promoting hormones and are biosynthesized in the young tissues to promote plant growth and development [32]. BR biosynthesis defect resulted in delayed plant growth, rounded rosette leaves and the increased the blade base angle, suggesting BR is involved in the vegetative phase change of plants [34,36,50]. However, the molecular mechanism by which BR regulates the vegetative phase change remains unclear. In this study, we showed that BZR1 interacted with SPL9 to integrate the age and BR signaling pathways to regulate the vegetative phase change. Mutation of BR receptor *BRI1* resulted in the significant reduced expression levels of *SPL9*, *SPL10* and *SPL15*, which prolonged the vegetative phase. The round rosette leaves of BR-deficient mutant *det2-1* was suppressed by the *pSPL9:rSPL9*, suggesting SPL9 regulate the vegetative growth of plants downstream of BR signaling. In addition, SPL9 directly interacted with BZR1 to promote the downstream gene expression. These results indicated that BR promotes the vegetative phase change through dual regulation of activity of SPL9 at the transcriptional and post-transcriptional levels.

MiR156 and its target gene SPL transcription factors are reported to regulate a wide range of plant growth and development by modulating the biosynthesis and signal transduction of plant hormones [9,10]. DELLA proteins, the key repressors of Gibberellin acid (GA) signaling pathway, interact with SPL9 to interfere SPL9 transcriptional activity and consequently delay floral transition and inhibit axillary meristem initiation [30,51]. In common wheat (*Triticum aestivum*), *TaSPL8* knock-out mutants displayed the erect leaves and increased spike number in high planting density. TaSPL8 directly binds to the promoters of BR biosynthesis gene *CYP90D2* and *AUXIN RESPONSE FACTOR* to induce their expression, suggesting TaSPL8 might increase lamina joint through auxin signaling and BR biosynthesis [52]. Here, in this study, we showed that the dwarf phenotypes of *det2-1* and *bri1-301* were partially suppressed by the *pSPL9:rSPL9* transgenic plants. The activation by expression *rSPL9* or *MIM156* resulted in the increased BR sensitivity and overexpression of miR156B partially suppressed the PPZ resistance of *bzr1-1D*. The *spl9*/*spl15* mutants displayed the BR response similar to that of wild type, which may be due to the functional redundancy of SPL gene family. BR induces the expression of several *SPL* genes. BZR1, the master regulator of BR signaling pathway, interacts with SPL9 to coordinately regulate the downstream gene expression. These results indicated that SPL9 is a positive regulator of BR signaling pathway, and work together with BZR1 to promote cell elongation and vegetative phase change in Arabidopsis.

## 4. Materials and Methods

### 4.1. Plant Materials and Growth Conditions

*Arabidopsis thaliana* Col-0 accession was the wild-type used as control for all phenotypic comparisons. Plants for general growth and seed harvesting were grown in the green house with 16-h light/8-h dark photoperiod at 22–24 °C. Arabidopsis lines used in this research include: *miR156B-Ox*, *MIM156*, *pSPL9:rSPL9* [9,28]; *det2-1*, *bri1-116*, *bir1-301*, *DWF4-Ox*, *bin2-1* [43,48,53,54,55]; *pSPL9:rSPL9/det2-1*, *pSPL9:rSPL9/bri1-301* (generated by *pSPL9:rSPL9* crossing with *det2-1* or *bri1-301*). For hypocotyl length, leaf phase (width/length), blade base angle measurement, the seedlings were scanned by aK708 scanner (BenQ, Shanghai, China), and measured using ImageJ software (National Institute of Mental Health, Bethesda, MD, USA). For rosette leaf numbers measurement, plants were grown under short-day conditions (8 h light/16 h dark) until flowering.

### 4.2. Plasmid and Transgenic Plants

Full-length cDNA of *SPL9* without stop codon were amplified by PCR and cloned into pENTR TM /SD/D-TOPO TM vectors (Thermo Fisher, Waltham, MA, USA) and then recombined with destination vector pGAL4BDGW (GAL4BD-X), pDEST15 (N-GST), and pX-nYFP (*p35S::X-nYFP*). For *bzr1-1D-Ox* transgenic plants, the constructs of *bzr1-1D* were performed using the quick-change site-directed mutagenesis kit (Strategene), and then recombined with destination vector pX-YFP (*p35S::bzr1-1D-YFP*), finally, introduced into *Agrobacterium tumefaciens* (strain GV3101), and transformed into Col-0 plants by the floral dipping method.

### 4.3. In Vitro Pull-Down Assays

Glutathione beads containing 1 µg of GST-SPL9 were incubated with 1 µg MBP or MBP- BZR1, which were purified from bacteria, in incubation buffer (20 mM Tris-HCl pH 7.5, 100 mM NaCl, 1 mM EDTA) at 4 °C for 1 h, and the beads were washed 10 times with wash buffer (20 mM Tris-HCl pH 7.5, 300 mM NaCl, 0.5% TritonX-100, 1 mM EDTA). The beads binding proteins complex were eluted with 2 × SDS sample buffer, and separated on 8% SDS-PAGE gels, then analyzed with anti-MBP (NEB, 1:5000 dilution) and anti-GST (Santa Cruz Biotechnology, Dallas, TX, USA, 1:5000 dilution) antibodies.

### 4.4. Bimolecular Fluorescence Complementation Assays

Full-length coding sequences of *SPL9* and *BZR1* were fused in-frame with the N-terminal of YFP and C-terminal of YFP, respectively. Agrobacterial suspensions containing SPL9-nYFP or BZR1-cYFP constructs were injected into tobacco (*Nicotiana tabacum*) leaves simultaneously. The transfected plants were cultivated in the greenhouse for at least 36 h at 22 °C, and then were used to analyze fluorescent signals using an LSM700 laser scanning confocal microscope (Zeiss, Oberkochen, Germany).

### 4.5. RNA Extraction, Reverse Transcription and Real-Time PCR

Total RNA was extracted from wild type and various mutants grown on ½ MS medium with 1% sucrose under constant light for 7 days using Trizol RNA extraction kit (TransGen Biotech, Beijing, China). First-strand cDNA were synthesized using RevertAid reverse transcriptase (Thermo Fisher, Waltham, MA, USA). Quantitative PCR analyses were performed on a CFX connect real-time PCR detection system (Bio-Rad, Hercules, CA, USA) using a SYBR green reagent (Roche, Basel, Switzerland) with gene-specific primers.

### 4.6. Transient Gene Expression Assays

Protoplast isolation and PEG transformation was carried out as described previously [56,57]. Protoplasts were harvested by centrifugation and lysed in 100 µL of passive lysis buffer (Promega, Madison, WI, USA). Firefly and Renilla (as internal standard) luciferase activities were measured using a dual-luciferase reporter kit (Promega, Madison, WI, USA).

## 5. Conclusions

In summary, we found that BR promotes the vegetative phase change by regulating SPL9 activity at both transcriptional and post-transcriptional levels. BR-deficient or -insensitive mutants displayed the round leaf and reduced leaf blade base angle, whereas *BRI1-Ox* and *DWF4-Ox* showed the slender leaf and increased leaf base angle. BR induces the expression of *SPLs*. SPL9 interacted with BZR1 to coordinately regulate the downstream gene expression. These results demonstrated that SPL9 regulates cell elongation and vegetative phase change downstream of developmental and hormonal signals.

## Figures and Tables

**Figure 1 ijms-22-10415-f001:**
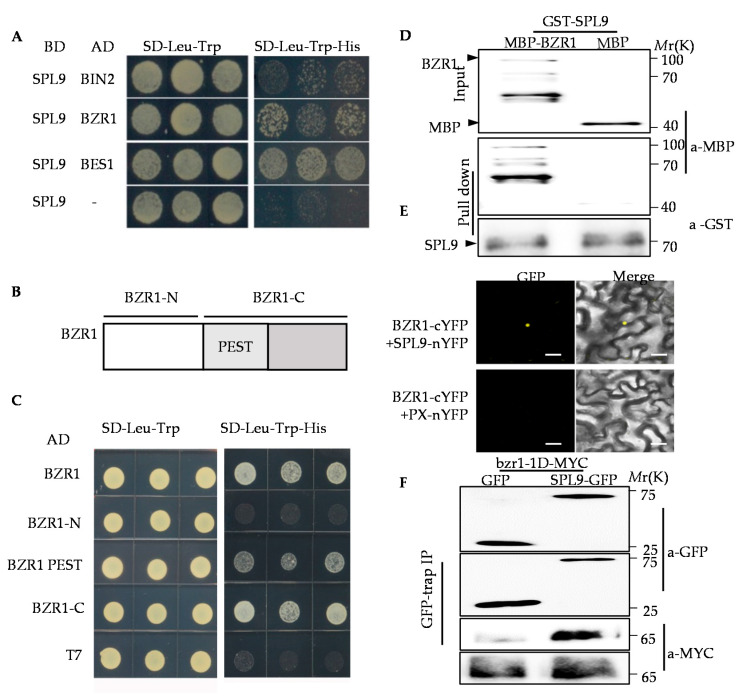
BZR1 interacts with SPL9 in vitro and in vivo. (**A**) SPL9 interacted with BZR1, BES1, and BIN2 in yeast. AD means activation domain; BD means DNA binding domain; SD means Synthetic dextrose minimal medium. (**B**) A diagram of the structure of BZR1. (**C**) The C-terminal of BZR1 was required for the interaction between BZR1 and SPL9 in yeast. PEST means the PEST domain. (**D**) Maltose binding protein (MBP) or MBP-BRASSINAZOLE RESISTANT 1 (BZR1) protein were incubated with glutathione-agarose beads combining glutathione S transferase (GST)-SQUAMOSA PROMOTER BINDING PROTEIN-LIKE9 (SPL9) and then eluted and analyzed by anti-MBP and anti-GST immunoblotting. (**E**) BiFC assays showed the interaction between BZR1 and SPL9 in plants. GFP means the green fluorescent protein. (**F**) BZR1 interacts with SPL9 in vivo. Immunoprecipitation (IP) was performed using Arabidopsis protoplast co-expressing *35S:GFP* and *35S:bzr1-1D-MYC* or co-expressing *35S:SPL9-GFP* and *35S:bzr1-1D-MYC.* The coimmunoprecipitation experiments were performed using GFP-Trap agarose beads, and the immunoblots were probed with anti-Myc or anti-YFP antibodies.

**Figure 2 ijms-22-10415-f002:**
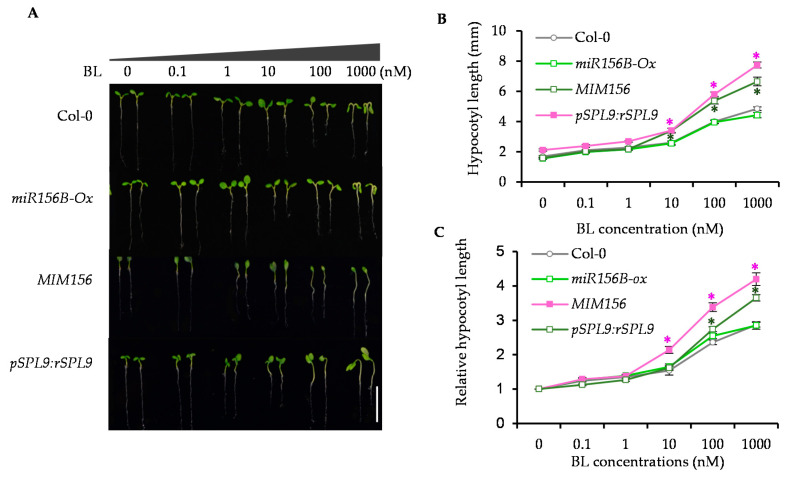
The *MIM156* and *pSPL9:SPL9* transgenic plants displayed the increased BR sensitivity on hypocotyl elongation. (**A**) Representative images of seven-day-old seedlings of wild-type and *miR156B-Ox*, *MIM156* and *pSPL9:rSPL9* grown on the ½ MS medium with different concentrations of brassinolide (BL). (**B**,**C**) Activated SPL9 by expressing the mimic *miR156* or the miR156-resistant form of *SPL9* increased the BR response. The hypocotyl lengths were measured from at least 30 plants. Relative hypocotyl length were average of 30 plants and normalized to the untreated plants. Error bars represent standard deviation. * *p* < 0.05.

**Figure 3 ijms-22-10415-f003:**
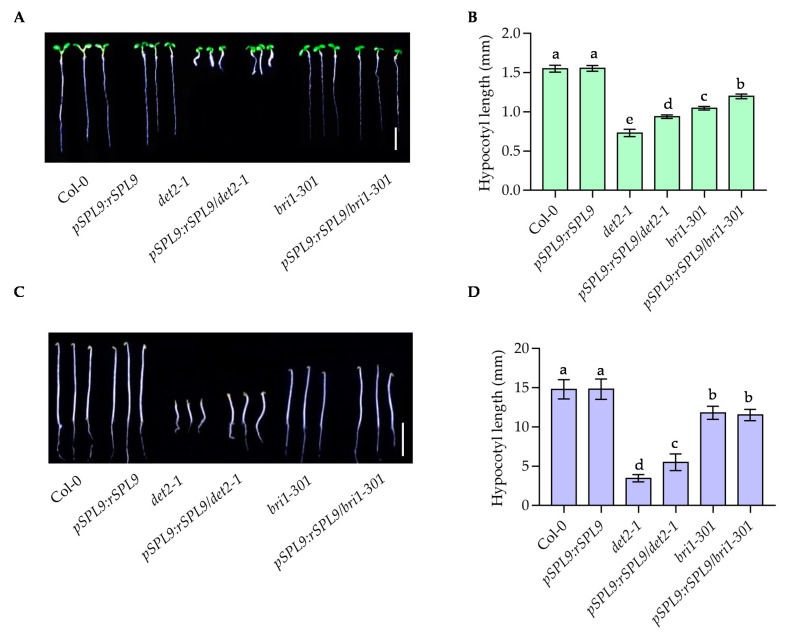
The dwarf phenotypes of *det2-1* and *bri1-301* were partially suppressed by the *pSPL9:rSPL9* transgenic plants. (**A**,**B**) Seedlings of wild type and indicated plants were grown under constant light for 7 days. Hypocotyl lengths were measured from at least 30 plants. Error bars represent standard deviation. Different letters above the bars indicated statistically significant differences between the samples (ANOVA analysis followed by Uncorrected Fisher’s LSD (Least-significant difference) multiple comparisons test, *p* < 0.05). (**C**,**D**) Seedlings of wild type and indicated plants were grown in the dark for 7 days. Hypocotyl lengths were measured from at least 30 plants. Error bars represent standard deviation. Different letters above the bars indicated statistically significant differences between the samples (ANOVA analysis followed by Uncorrected Fisher’s LSD multiple comparisons test, *p* < 0.05).

**Figure 4 ijms-22-10415-f004:**
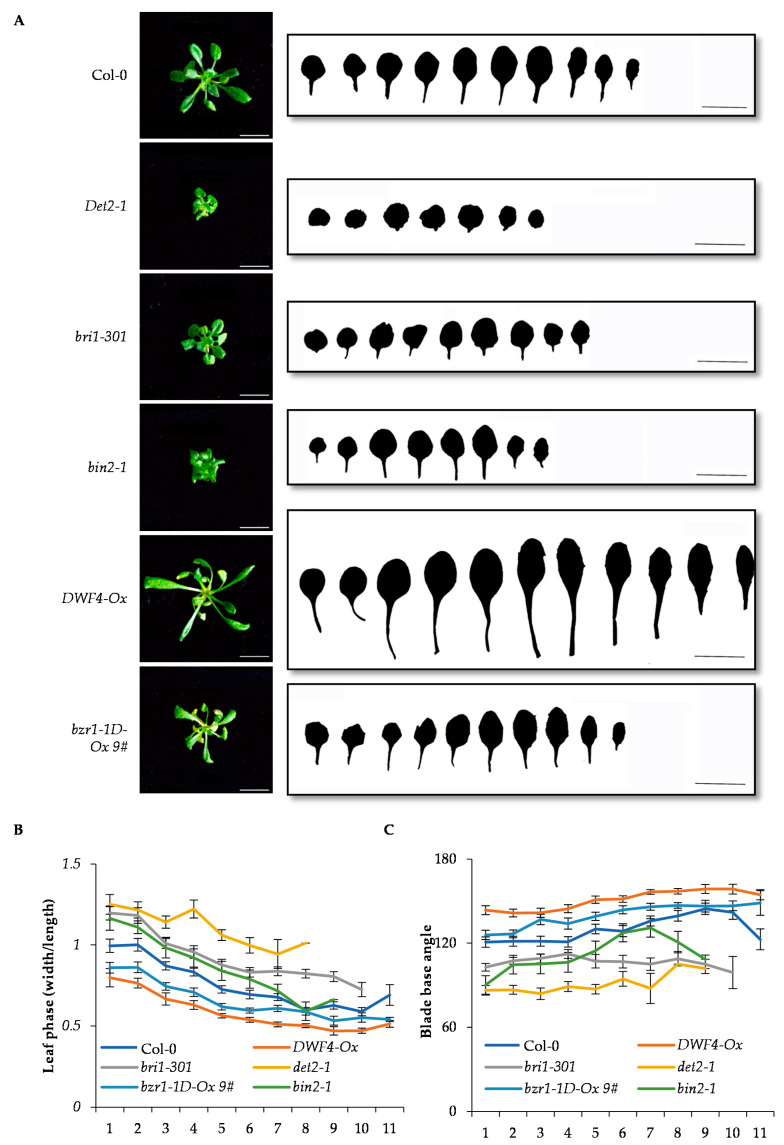
Brassinosteroid (BR) promotes vegetative phase change of plants. (**A**) The shape phenotypes of full expanded rosette leaves of wild type and different BR-related mutants. (**B**) The ratio of leaf width to leaf length of full expanded rosette leaves of wild type and different BR-related mutants. The leaf width and leaf length were measure from at least 30 plants. Error bars represent standard deviation. The x axis indicates the number of leaves. (**C**) The base angle of full expanded rosette leaves of wild type and different BR-related mutants. The leaf base angle were measure from at least 30 plants. Error bars represent standard deviation. The x axis indicates the number of leaves.

**Figure 5 ijms-22-10415-f005:**
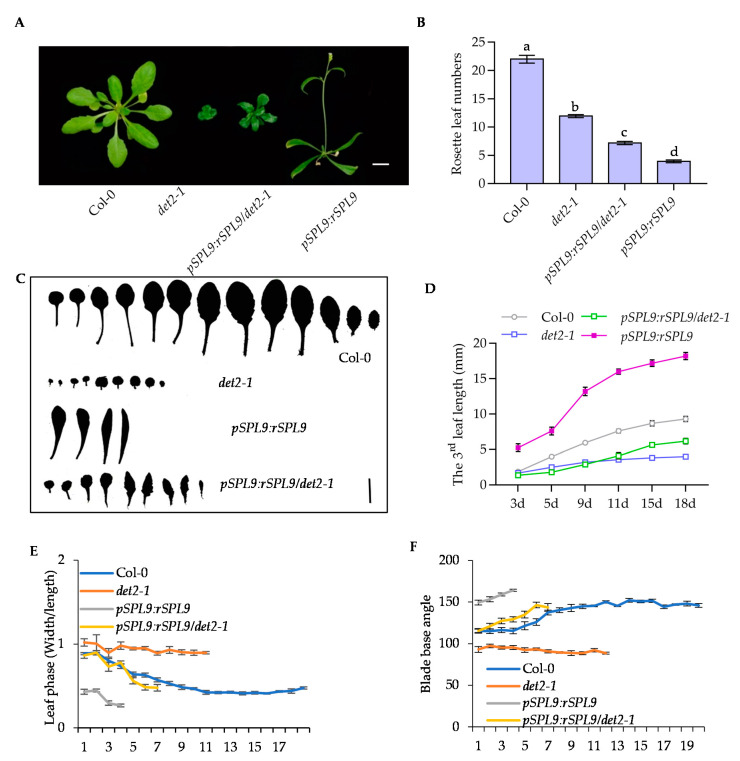
The *pSPL9:rSPL9* suppressed the delayed vegetative phase change of *det2-1* mutant. (**A**) The growth phenotype of 5-week-old wild type, *det2-1*, *pSPL9:rSPL9* and *pSPL9:rSPL9*/*det2-1* plants. (**B**) The rosette leaf numbers of wild type, *det2-1*, *pSPL9:rSPL9* and *pSPL9:rSPL9*/*det2-1* after flowering. Error bars represent standard deviation (*n* = 10). Different letters above the bars indicated statistically significant differences between the samples (ANOVA analysis followed by Uncorrected Fisher’s LSD multiple comparisons test, *p* < 0.05). (**C**) The shape phenotypes of full expanded rosette leaves of wild type, *det2-1*, *pSPL9:rSPL9* and *pSPL9:rSPL9*/*det2-1*. (**D**) The 3rd leaf length of wild type, *det2-1*, *pSPL9:rSPL9* and *pSPL9:rSPL9*/*det2-1* after initiation different days. The leaf lengths were measure from at least 30 plants. Error bars represent standard deviation. (**E**) The ratio of leaf width to leaf length of full expanded rosette leaves of wild type and indicated plants. The leaf width and leaf length were measure from at least 30 plants. Error bars represent standard deviation. (**F**) The base angle of full expanded rosette leaves of wild type and indicated plants. The leaf base angle were measure from at least 30 plants. Error bars represent standard deviation.

**Figure 6 ijms-22-10415-f006:**
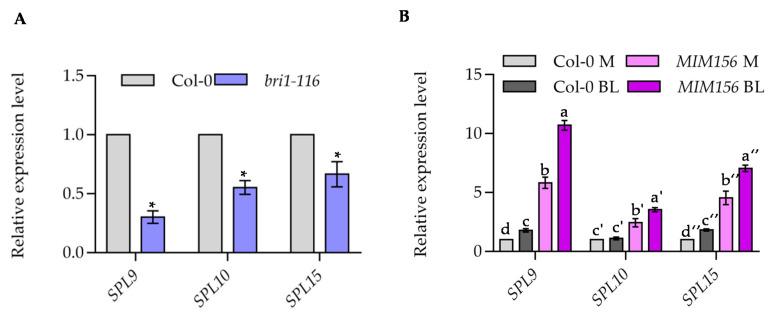
BR induces the expression of *SPL9*, *SPL10* and *SPL15*. (**A**) Quantitative RT-PCR analysis the expression of *SPL9*, *SPL10* and *SPL15* in wild type and *bri1-116* mutants. Seedlings of Col-0 and *bri1-116* were grown on ½ MS medium under constant light for 7 days. *PP2A* gene was analyzed as an internal control. Error bars represent standard deviation of three independent experiments. Asterisk between bars indicated statistically significant differences between the samples (Student t test, * *p* < 0.05). (**B**) BR increased the transcript of *SPL9*, *SPL10* and *SPL15*. Seedlings of wild type and *MIM156* were grown on ½ MS medium for 7 days, and then treated with 100 nM BL for 3 h. *PP2A* gene was analyzed as an internal control. Error bars represent standard deviation of three independent experiments. Different letters above the bars indicated statistically significant differences between the samples (ANOVA analysis followed by Uncorrected Fisher’s LSD multiple comparisons test, *p* < 0.05).

**Figure 7 ijms-22-10415-f007:**
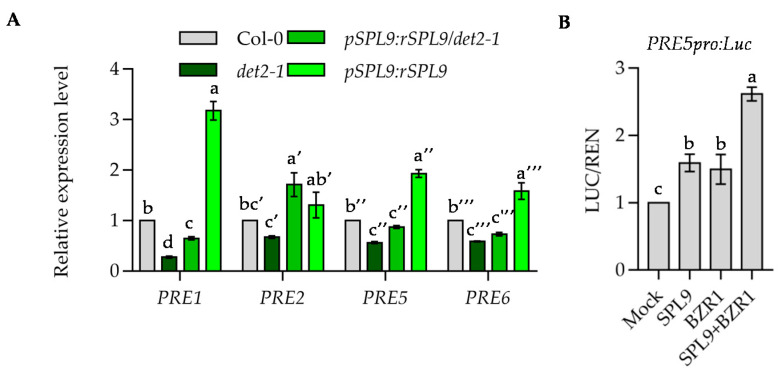
BZR1 and SPL9 cooperatively regulate the downstream gene expression. (**A**) Quantitative RT-PCR analysis the expression of members of PRE family in wild type and indicated plants. Seedlings of wild type, *det2-1*, *pSPL9:rSPL9* and *pSPL9:rSPL9*/*det2-1* were grown on ½ MS medium under constant light for 7 days. *PP2A* gene was analyzed as an internal control. Error bars represent standard deviation of three independent experiments. Different letters above the bars indicated statistically significant differences between the samples (ANOVA analysis followed by Uncorrected Fisher’s LSD multiple comparisons test, *p* < 0.05). (**B**) Transient assays show BZR1 and SPL9 cooperative activation of the *pPRE5:LUC* reporter gene. The construct containing *pPRE5:LUC* (luciferase) and *35S:REN* (renilla luciferase) and constructs overexpressing the indicated effecters were transfected to Arabidopsis protoplasts simultaneously. The LUC activity was normalized to REN. Error bars indicate SD of three biological repeats. Different letters above the bars indicated statistically significant differences between the samples (ANOVA analysis followed by Uncorrected Fisher’s LSD multiple comparisons test, *p* < 0.05).

## Data Availability

Not applicable.

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
