# Peer review of "BZR1 Physically Interacts with SPL9 to Regulate the Vegetative Phase Change and Cell Elongation in Arabidopsis"

_ijms, 2021, doi:10.3390/ijms221910415_

Round 1
Reviewer 1 Report
In this work, Wang et al show that BZR1 (a key transcription factor in BR signaling) directly interacts with SPL9 (a positive regulator in the vegetative phase transition). Genetic analyses show that SPL9 enhances BR-mediated cell elongation, and BR signaling promotes vegetative phase transition. They also show that BR signaling increases the expression of some SPL genes including SPL9, and BZR1 and SPL9 cooperatively regulates the expression of PRE genes. Overall, the results are interesting, but major concerns need to be addressed.
1. In vivo interaction between BZR1 and SPL9 should be confirmed by co-IP assays because of high false-positive rates in BiFC assays (see the following article, Horstman et al., (2014, Int J Mol Sci).
2. Genetic results with loss-of-function mutants should be included. Authors need to include spl9 loss-of-function mutants in the assays described in Fig 2. and 7. It is likely that spl9 mutants do not exhibit obvious phenotypes, probably due to genetic redundancy. Even if this is the case, the results should be presented along with the results with gain-of-function mutants (e.g., pSPL9:rSLP9), and potential explanation for the lack of phenotypes should be addressed in the Discussion section. (e.g., BZR1 interacts with other SPLs)
3. Authors used the ratio of leaf width to length as a proxy for the leaf phase. However, the altered ratio could be simply due to the effects of BR on leaf shape. Therefore, they need to use another marker (e.g., the appearance of abaxial trichomes) to complement their results.
4. Author argue that BR promotes vegetative phase transition. However, the det2 mutants seem to flower earlier than WT in Fig. 5B. How is the early flowering phenotype of the det2 mutant explained if BR promotes vegetative phase transition?
Minor point(s)
1. Statistical analysis is missing in Fig. 2(B, C)
2. a letter is missing in Fig. 6B (SLP15)
Author Response
Response to reviewer’s comments:
In this work, Wang et al show that BZR1 (a key transcription factor in BR signaling) directly interacts with SPL9 (a positive regulator in the vegetative phase transition). 8Genetic analyses show that SPL9 enhances BR-mediated cell elongation, and BR signaling promotes vegetative phase transition. They also show that BR signaling increases the expression of some SPL genes including SPL9, and BZR1 and SPL9 cooperatively regulates the expression of PRE genes. Overall, the results are interesting, but major concerns need to be addressed.
- In vivo interaction between BZR1 and SPL9 should be confirmed by co-IP assays because of high false-positive rates in BiFC assays (see the following article, Horstman et al., (2014, Int J Mol Sci).
Response:Thank you for pointing this out. To further verify the interaction between BZR1 and SPL9 in plants, we have performed Co-IP assay using YFP-trap beads and Arabidopsis protoplast,in which co-expressing 35S:GFP and 35S:bzr1-1D-MYC or co-expressing 35S:SPL9-GFP and 35S:bzr1-1D-MYC. The results confirmed that BZR1 interacted with SPL9 in plants.
Figure 1. BZR1 interacts with SPL9 in vitro and in vivo.
(F) BZR1 interacts with SPL9 in vivo. Immunoprecipitation was performed using Arabidopsis protoplast co-expressing 35S:GFP and 35S:bzr1-1D-MYC or co-expressing 35S:SPL9-GFP and 35S:bzr1-1D-MYC. The immunoblots were probed with anti-Myc or anti-YFP antibodies.
- Genetic results with loss-of-function mutants should be included. Authors need to include spl9 loss-of-function mutants in the assays described in Fig 2. and 7. It is likely that spl9 mutants do not exhibit obvious phenotypes, probably due to genetic redundancy. Even if this is the case, the results should be presented along with the results with gain-of-function mutants (e.g., pSPL9:rSLP9), and potential explanation for the lack of phenotypes should be addressed in the Discussion section. (e.g., BZR1 interacts with other SPLs)
Response:Thank you for pointing this out. We have analyzed the BR response on hypocotyl elongation of spl9 spl15 loss-of-function mutant. The spl9 spl15 displayed the similar BR response to wild-type plants. Additionally, there are no difference of PREs expression level between spl9 spl15 and wild-type plants. The SPL genes targeted by miR156 can be grouped into four major clades: SPL3/SPL4/SPL5, SPL2/SPL10/SPL11, SPL9/SPL15, SPL6/SPL13[1]. Previous study has reported that SPL3, SPL9 and SPL10/SPL11 have overlapping, but distinct functions in vegetative development[2]. So, loss-of-function mutations in SPL9 and SPL15 had no obvious hypocotyl length phenotype, presumably because BZR1 interacts with other SPLs and there is functional redundancy in SPL genes.
Figure S1. The BR response of spl9/spl15 on hypocotyl elongation.
(A) Representative images of seven-day-old seedlings of wild-type, and spl9/spl15 grown on the ½ MS medium with different concentrations of BL. (B) and (C) The hypocotyl lengths were measured from at least 30 plants. Relative hypocotyl length was average of 30 plants and normalized to the untreated plants. Error bars represent standard deviation. (D) Quantitative RT-PCR analysis the expression of members of PRE family in wild type, spl9 and spl9spl15. Seedlings of wild type and indicated plants were grown on ½ MS medium under constant light for 7 days. PP2A gene was analyzed as an internal control. Error bars represent standard deviation of three independent experiments.
References:
- Guo, A.Y.; Zhu, Q.H.; Gu, X.; Ge, S.; Yang, J.; Luo, J. Genome-wide identification and evolutionary analysis of the plant specific SBP-box transcription factor family. Gene 2008, 418, 1-8, doi:10.1016/j.gene.2008.03.016.
- Wu, G.; Park, M.Y.; Conway, S.R.; Wang, J.W.; Weigel, D.; Poethig, R.S. The sequential action of miR156 and miR172 regulates developmental timing in Arabidopsis. Cell 2009, 138, 750-759, doi:10.1016/j.cell.2009.06.031.
- Authors used the ratio of leaf width to length as a proxy for the leaf phase. However, the altered ratio could be simply due to the effects of BR on leaf shape. Therefore, they need to use another marker (e.g., the appearance of abaxial trichomes) to complement their results.
Response:Thank you for pointing this out. We have analyzed the 1st leaf with abaxial trichomes of BR related mutants and genetic materials. The results showed that Col-0 appeared the abaxial trichome in the 7th rosette leaf, while bri1-301 appeared the abaxial trichome in the 9th leaf. Furthermore, we found bri1-301 significantly delayed the appearance of abaxial trichomes in pSPL9:rSPL9 plant. These results suggested BR signaling promoted the vegetative phase change.
Figure S5. bri1-301 can partly rescue the less abaxial trichome phenotype of pSPL9:rSPL9.
The growth phenotype of 5-week-old wild type, bri1-301, pSPL9:rSPL9 and pSPL9:rSPL9/ bri1-301 plants. The number of 1st leaf with abaxial trichome of indicated plant were calculated from at least 10 plants. Error bars represent standard deviation.
- Author argue that BR promotes vegetative phase transition. However, the det2 mutants seem to flower earlier than WT in Fig. 5B. How is the early flowering phenotype of the det2 mutant explained if BR promotes vegetative phase transition?
Response:Thank you for pointing this out. Although BR-deficient mutant det2-1 displayed the fewer rosettes leaves at flowering than wild type, det2-1 mutants need more time to bolting. The det2-1 mutants exhibited delayed flowering time by at least 10 days compared with the wild type under long day condition[1-3]. The time of vegetative phase is much longer in det2-1 mutant than that in wild type, suggesting BR positively regulates vegetative phase transition in Arabidopsis.
References:
- Chory, J.; Nagpal, P.; Peto, C.A. Phenotypic and Genetic Analysis of det2, a New Mutant That Affects Light-Regulated Seedling Development in Arabidopsis. Plant Cell 1991, 3, 445-459, doi:10.1105/tpc.3.5.445.
- Li, Z.; Ou, Y.; Zhang, Z.; Li, J.; He, Y. Brassinosteroid Signaling Recruits Histone 3 Lysine-27 Demethylation Activity to FLOWERING LOCUS C Chromatin to Inhibit the Floral Transition in Arabidopsis. Mol Plant 2018, 11, 1135-1146, doi:10.1016/j.molp.2018.06.007.
- Hong, J.; Lee, H.; Lee, J.; Kim, H.; Ryu, H. ABSCISIC ACID-INSENSITIVE 3 is involved in brassinosteroid-mediated regulation of flowering in plants. Plant Physiol Biochem 2019, 139, 207-214, doi:10.1016/j.plaphy.2019.03.022.
Minor point(s)
- Statistical analysis is missing in Fig. 2(B, C)
Response:Thank you for pointing this out, we have added it.
- a letter is missing in Fig. 6B (SLP15)
Response:Thank you for pointing this out, we have changed it.

Reviewer 2 Report
Manuscript ID: ijms-1338428
In this manuscript, the authors used molecular biology methods and analyzed tons of genetic interactions, trying to figure out the role of SPL9 protein in brassinosteroid regulatory pathways, and in this way, to explain the observed mutants and transgenic plants phenotypes in Arabidopsis vegetative phase change and cell elongation. However, a major flaw of logic makes the conclusion of this manuscript unreliable.
This manuscript started with an important Y2H library screening result using BZR1 as the bait, pointing out the physical interaction between BZR1 and SPL9 protein is the key. This is the basis of the entire hypothesis that SPL9 might be important in BR regulatory pathway. However, throughout the whole manuscript, the authors ignored this interaction, and moved on to discuss the potential role of SPL9 in BR responses, with an emphasis on the known regulation between miRNA156 and SPL proteins. Although the authors wrote clearly in the title of this manuscript, they failed to present how the physical interactions with SPL9 makes BZR1 important in the observed phenotypes. Here, BZR1 should be the core subject to study, isn’t it? Does BZR1 require SPL9 interaction to carry out BZR1’s normal functions in the BR regulatory pathway? Does this interaction between BZR1 and SPL9 take place in the nucleus? Both being transcription factors according to previous knowledge, does BZR1 and SPL9 target the same genes together? If so, are they competitors for regulating the same genes, or like the authors showed with the PRE5 promoter in Figure 7, either BZR1 or SPL9 would induce its expression in the protoplasts, but the expression induction is even higher when both BZR1 and SPL9 are present, but how does this happen? Does this additive effect rely on the physical interactions between BZR1 and SPL9?? You see, questions like this cannot be explained until the authors dive deeper into the molecular mechanisms behind their observation.
The physical interactions as the authors presented in Figure 1 looks solid, although it would be better to have a co-IP result instead of an in vitro pull down experiment. This result is important, please be very careful with the conclusion. If the authors decided not to further investigate the impact of SPL9 interaction on BZR1, they would probably have to leave this part out of the manuscript, and build a story solely based on SPL9.
Author Response
Response to reviewer’s comments:
In this manuscript, the authors used molecular biology methods and analyzed tons of genetic interactions, trying to figure out the role of SPL9 protein in brassinosteroid regulatory pathways, and in this way, to explain the observed mutants and transgenic plants phenotypes in Arabidopsis vegetative phase change and cell elongation. However, a major flaw of logic makes the conclusion of this manuscript unreliable.
This manuscript started with an important Y2H library screening result using BZR1 as the bait, pointing out the physical interaction between BZR1 and SPL9 protein is the key. This is the basis of the entire hypothesis that SPL9 might be important in BR regulatory pathway. However, throughout the whole manuscript, the authors ignored this interaction, and moved on to discuss the potential role of SPL9 in BR responses, with an emphasis on the known regulation between miRNA156 and SPL proteins. Although the authors wrote clearly in the title of this manuscript, they failed to present how the physical interactions with SPL9 makes BZR1 important in the observed phenotypes. Here, BZR1 should be the core subject to study, isn’t it? Does BZR1 require SPL9 interaction to carry out BZR1’s normal functions in the BR regulatory pathway?
Response:Thank you for this excellent suggestion. To investigate the effects of SPL9 on the functions of BZR1, we analyzed the growth phenotypes of Col-0, miR156-Ox, bzr1-1D and bzr1-1D/miR156-Ox that were grown on the ½ MS medium with or without 2 µM PPZ for 3 weeks. The results showed that bzr1-1D is resistant to PPZ, but such resistance of bzr1-1D were significantly reduced by miR156 overexpression. This result suggested miR156-SPL module plays an important role for the functions of BZR1 on cell elongation.
Figure S3. MiR156 overexpression partially suppressed the PPZ resistance of bzr1-1D.
(A) The representative image of 3-week-old wild type, miR156B-Ox, bzr1-1D and bzr1-1D/miR156B-Ox plants. (B) The leaf area of 3rd rosette leaf of wild type and indicated plants. Error bars represent standard deviation (n=10).
Does this interaction between BZR1 and SPL9 take place in the nucleus?
Response: The BiFC experiments showed the fluorescent signals of YFP were observed in the nucleus of epidermal cells of tobacco leaves that co-expressed BZR1-cYFP and SPL9-nYFP. This result suggested that BZR1 interacts with SPL9 in the nucleus of plants.
Both being transcription factors according to previous knowledge, does BZR1 and SPL9 target the same genes together? If so, are they competitors for regulating the same genes, or like the authors showed with the PRE5 promoter in Figure 7, either BZR1 or SPL9 would induce its expression in the protoplasts, but the expression induction is even higher when both BZR1 and SPL9 are present, but how does this happen? Does this additive effect rely on the physical interactions between BZR1 and SPL9?? You see, questions like this cannot be explained until the authors dive deeper into the molecular mechanisms behind their observation.
Response:Thank you for pointing this out. The transient expression assay showed that BZR1 and SPL9 could induce the expression of PRE5, but the expression of PRE5 was much higher in the presence of both BZR1 and SPL9 (Fig. 7B). SPL9 induces the expression of PRE1, PRE5 and PRE6 in plants, but such induction of SPL9 were almost abolished in det2-1 mutant background (Fig. 7A). In addition, our genetic analysis showed that det2-1 partially suppressed the growth phenotype of pSPL9:rSPL9, and miR156 overexpression reduced the PPZ sensitivity of bzr1-1D. These results suggested BZR1 interacted with SPL9 to enhance the transcriptional activity each other, then promoting the downstream gene expression and plant growth.
The physical interactions as the authors presented in Figure 1 looks solid, although it would be better to have a co-IP result instead of an in vitro pull down experiment. This result is important, please be very careful with the conclusion. If the authors decided not to further investigate the impact of SPL9 interaction on BZR1, they would probably have to leave this part out of the manuscript, and build a story solely based on SPL9.
Response:Thank you for pointing this out. To further verify interaction between BZR1 and SPL9 in vivo, we have performed Co-IP assay using YFP-trap beads and Arabidopsis protoplast co-expressing 35S-GFP and 35S:bzr1-1D-MYC or co-expressing 35S:SPL9-GFP and 35S:bzr1-1D-MYC. The results confirmed that BZR1 interacted with SPL9 in plants.
Figure 1. BZR1 interacts with SPL9 in vitro and in vivo.
(F) BZR1 interacts with SPL9 in vivo. Immunoprecipitation was performed using Arabidopsis protoplast co-expressing 35S:GFP and 35S:bzr1-1D-MYC or co-expressing 35S:SPL9-GFP and 35S:bzr1-1D-MYC. The immunoblots were probed with anti-Myc or anti-YFP antibodies.

Round 2
Reviewer 2 Report
The authors addressed my questions in the previous review report, and made modifications to the manuscript that imporoved its overall quality. But still, the authors failed to clearly present the direct molecular mechanism between the physical interactions between BZR1 and SPL9, and the fact that BR affected phenotypes that are known to be regulated by miR156-SPL9. I understand that both BR pathways and miR156-SPL module regulate a lot of stuff and are both regulated or have interactions with a lot of other genetic and epigeneic regulation networks, so it is very difficult to address this one-on-one interaction and the causal effect that this interaction accounts for, using only genetic analysis. I still do not agree with the way the authors organize and present the valuable data that they collected, and apparently the authors do not agree with my opinions either. I appreciate the hard work that was performed by the authors, and I still suggest that the authors rewrite and weaken the way they describe the physical interactions between BZR1 and SPL9 in the overall effect of BR in miR156-SPL9-related plant growth. Also, please carefully go through the manuscript and correct minor typo errors here and there, if the editors decided to accept the manuscript. For example, Line 92, a space is missing in "increasethe".
Author Response
Response to reviewer’s comment:
Reviewer 2
The authors addressed my questions in the previous review report, and made modifications to the manuscript that improved its overall quality. But still, the authors failed to clearly present the direct molecular mechanism between the physical interactions between BZR1 and SPL9, and the fact that BR affected phenotypes that are known to be regulated by miR156-SPL9. I understand that both BR pathways and miR156-SPL module regulate a lot of stuff and are both regulated or have interactions with a lot of other genetic and epigenetic regulation networks, so it is very difficult to address this one-on-one interaction and the causal effect that this interaction accounts for, using only genetic analysis. I still do not agree with the way the authors organize and present the valuable data that they collected, and apparently the authors do not agree with my opinions either. I appreciate the hard work that was performed by the authors, and I still suggest that the authors rewrite and weaken the way they describe the physical interactions between BZR1 and SPL9 in the overall effect of BR in miR156-SPL9-related plant growth.
Response:Thank you for pointing this out. Our genetic analysis showed that the dwarf phenotypes of det2-1 and bri1-301 were partially suppressed by the pSPL9:rSPL9 transgenic plants. The activation by expression rSPL9 or MIM156 resulted in the increased BR sensitivity and overexpression of miR156 partially suppressed the PPZ resistance of bzr1-1D. The expression levels of SPL9, SPL10 and SPL15 were significantly induced by BR treatment, but reduced in bri1-116 mutant compared to wild-type plants. BZR1 interacted with SPL9 to cooperatively regulate the expression of downstream genes. Together, these genetic and biochemical results provided the strong evidence that SPL9 plays critical roles for the functions of BZR1 in cell elongation and vegetative phase transition.
Also, please carefully go through the manuscript and correct minor typo errors here and there, if the editors decided to accept the manuscript. For example, Line 92, a space is missing in "increasethe".
Response:Thank you for pointing this out. We have extensively revised the language of our manuscript to correct grammatical and typo errors.
